# Agreement and relationship between measures of absolute and relative intensity during walking: A systematic review with meta-regression

Ashley Warner[1], Natalie Vanicek[1], Amanda Benson[2], Tony Myers[3], Grant Abt[1] *

1 Department of Sport, Health and Exercise Science, University of Hull, Hull, United Kingdom, 2 School of Health Sciences, Swinburne University of Technology, Melbourne, Australia, 3 Department of Sport and Health, Newman University, Birmingham, United Kingdom

☯ These authors contributed equally to this work.

* g.abt@hull.ac.uk

## Abstract

### Introduction

A metabolic equivalent (MET) is one of the most common methods used to objectively quantify physical activity intensity. Although the MET provides an 'objective' measure, it does not account for inter-individual differences in cardiorespiratory fitness. In contrast, 'relative' measures of physical activity intensity, such as heart rate reserve (HRR), do account for cardiorespiratory fitness. The purpose of this systematic review with meta-regression was to compare measures of absolute and relative physical activity intensity collected during walking.

### Methods

A systematic search of four databases (SPORTDiscus, Medline, Academic Search Premier and CINAHL) was completed. Keyword searches were: (i) step* OR walk* OR strid* OR "physical activity"; (ii) absolute OR "absolute intensity" OR mets OR metabolic equivalent OR actigraph* OR acceleromet*; (iii) relative OR "relative intensity" OR "heart rate" OR "heart rate reserve" OR "VO$_2$ reserve" OR VO$_2$* OR "VO$_2$ uptake" OR HRmax* OR metmax. Categories (i) to (iii) were combined using 'AND;' with studies related to running excluded. A Bayesian regression was conducted to quantify the relationship between METs and %HRR, with Bayesian logistic regression conducted to examine the classification agreement between methods. A modified Downs and Black scale incorporating 13 questions relative to cross-sectional study design was used to assess quality and risk of bias in all included studies.

### Results

A total of 15 papers were included in the systematic review. A comparison of means between absolute (METs) and relative (%HRR, %HR$_{max}$, %VO$_2$R, %VO$_{2max}$, HR$_{index}$)

**Data Availability Statement:** The data underlying the raw agreement and meta-regression are available on request from the authors of the following studies: Sweegers MG, Buffart LM,

Huijsmans RJ, Konings IR, van Zweeden AA, Brug J, Chinapaw MJM, Altenburg TM. From accelerometer output to physical activity intensities in breast cancer patients. J Sci Med Sport. 2020 Feb;23(2):176-181. doi: 10.1016/j.jsams.2019.09.001. Gil-Rey E, Maldonado-Martín S, Palacios-Samper N, Gorostiaga EM. Objectively measured absolute and relative physical activity intensity levels in postmenopausal women. Eur J Sport Sci. 2019 May;19(4):539-548. doi: 10.1080/17461391.2018.1539528. Dos Anjos LA, Machado Jda M, Wahrlich V, De Vasconcellos MT, Caspersen CJ. Absolute and relative energy costs of walking in a Brazilian adult probability sample. Med Sci Sports Exerc. 2011 Nov;43(11):2211-8. doi: 10.1249/MSS.0b013e31821f5798.

**Funding:** The author(s) received no specific funding for this work.

**Competing interests:** The authors have declared that no competing interests exist.

**Abbreviations:** METs, metabolic equivalents; %HRR, percent heart rate reserve; $VO_2R$, oxygen consumption reserve; $\%VO_2R$, percent oxygen consumption reserve; $HR_{max}$, maximal heart rate; $\%HR_{max}$, percent maximal heart rate; $VO_{2peak}$, peak oxygen consumption; $\%VO_{2peak}$, percent peak oxygen consumption; $HR_{index}$, heart rate index; MVPA, moderate-to-vigorous physical activity; ACSM, American College of Sports Medicine.

values in 8 studies identified agreement in how intensity was classified (light, moderate or vigorous) in 60% of the trials. We received raw data from three authors, incorporating 3 studies and 290 participants. A Bayesian random intercept logistic regression was conducted to examine the agreement between relative and absolute intensity, showing agreement in 43% of all trials. Two studies had identical relative variables (%HRR) totalling 240 participants included in the Bayesian random intercept regression. The best performing model was a log-log regression, which showed that for every 1% increase in METs, %HRR increased by 1.12% (95% CI: 1.10–1.14). Specifically, the model predicts at the lower bound of absolute moderate intensity (3 METs), %HRR was estimated to be 33% (95%CI: 18–57) and at vigorous intensity (6 METs) %HRR was estimated to be 71% (38–100).

## Conclusion

This study highlights the discrepancies between absolute and relative measures of physical activity intensity during walking with large disagreement observed between methods and large variation in %HRR at a given MET. Consequently, health professionals should be aware of this lack of agreement between absolute and relative measures. Moreover, if we are to move towards a more individualised approach to exercise prescription and monitoring as advocated, relative intensity could be more highly prioritised.

## Introduction

Walking is a very popular form of physical activity at a population level [1]. It is low cost, accessible, and well-tolerated across age groups [2]. Given this popularity, walking is a key intervention for physical activity promotion [3]. Advocating walking (and physical activity in general) is important because it is well documented that physical activity reduces the risk of developing a range of chronic diseases [4, 5]. Accumulating a sufficient dose of physical activity is therefore important.

A key question for how the dose of physical activity is quantified is therefore how exercise intensity (referred to as intensity from now on) can be measured objectively. Generally, objective intensity can be measured via absolute or relative methods. Absolute intensity is a measure unrelated to the individual's cardiorespiratory fitness and is often measured via an accelerometer [6]. Relative intensity is a measure that is proportional to the cardiorespiratory fitness of the individual, and usually involves the measurement of heart rate or oxygen consumption. One of the most common absolute methods of objectively quantifying intensity is the metabolic equivalent (METs), which is a ratio of the metabolic cost induced by different types of exercise and intensity compared to the metabolic cost of sitting quietly [7]. The ability of METs to quantify intensity across a spectrum of physical activity, to a wide range of the population, has ensured its popularity among researchers and health and fitness professionals. For example, a walking cadence $\geq$100 steps.min$^{-1}$ has been recommended as sufficient to achieve moderate intensity walking as determined by a threshold of 3 METs [8].

There is, however, growing interest in relative methods of measuring intensity, allowing for more accurate, inter-individual prescription [9]. A relative method of measuring intensity is one that relates to one or more physiological characteristics of the individual, such as maximal and resting heart rates [10]. This relative (individualised) approach has widely been accepted in terms of scientific rigour but is growing in popularity in the general population. This is in

part due to the accessibility that wearable devices now offer in terms of heart rate measurement and real-time data collection that can be used to gauge intensity, duration and exercise modality [11, 12].

There are several methods employed to individualise intensity using relative methods. A number of variables recognised by the American College of Sports Medicine (ACSM) have sought to use measures of maximal physiological capacity, including maximal oxygen consumption ($VO_{2max}$), peak oxygen uptake ($VO_{2Peak}$), maximal heart rate ($HR_{max}$) and peak heart rate ($HR_{peak}$) to prescribe intensity [10]. These measures, usually derived from a maximal graded exercise test, can also be predicted using formulas based on age [13–16] and extrapolated from submaximal exercise tests [17–19] with good accuracy [20, 21]. Other relative methods of individualising intensity seek to incorporate both maximal and/or resting values as a means of regulating physical activity and exercise prescription based on fitness/health status [9, 22, 23]. Resting heart rate is an indicator of cardiorespiratory health, improved quality of life and improved life expectancy [18, 24]. Incorporating resting, alongside maximal values, using percentage heart rate reserve (%HRR) and percentage $VO_{2reserve}$ (%$VO_2$R), has the potential to improve accuracy of exercise and physical activity prescription when compared to using maximal data only [25, 26].

Given that intensity can be measured through both absolute and relative methods, it is important that we understand how these methods relate to each other and that we know the agreement between absolute and relative intensity measures. There is the potential for large disagreement, given that relative methods are based on the individual's cardiorespiratory fitness and/or other individualised measures. The agreement or disagreement between absolute and relative intensity methods has implications for physical activity and exercise monitoring and for how physical activity is promoted and measured in the general population, specifically when walking. Therefore, the purpose of this systematic review with meta-regression was to investigate the agreement and relationship between absolute (METs) and relative intensity methods during walking.

## Methods

### Protocol and registration

This systematic review with meta-regression was designed and written using the Preferred Reporting Items for Systematic Reviews and Meta-analyses (PRISMA) guidelines [27], and pre-registered on the Open Science Framework (https://osf.io/4md3z/?view_only=d3ffedd1738c4f2b858c8b4827c2e421) prior to conducting the first literature search.

### Eligibility criteria

Studies were only considered if they met the following inclusion criteria: published in English, peer-reviewed, focused on walking and included both absolute and relative intensity measures. To be eligible for inclusion, each study required a minimum of one of the ACSM [28] criterion measures of relative intensity, %HRR, %$VO_2$R, percent of heart rate max (%$HR_{max}$) or percent of maximal oxygen consumption (%$VO_{2max}$), together with the measurement of METs. We also included other relative measures including $HR_{index}$, lactate and ventilatory thresholds. The main population focus was adults aged 18–65 years old, apparently healthy and of any sex. Studies were excluded for the following reasons: participants were aged under 18 or over 65 years, full text was unavailable, the study involved animals, or participants had underlying health issues that may have impacted walking gait.

## Search strategy and data extraction

A systematic search of four databases (SPORTDiscus, Medline, Academic Search Premier and CINAHL) was completed from the earliest available date to September 2021 (Fig 1). Search terms consisted of activity (walking) and intensity (absolute and relative). First, keyword searches were conducted: (i) step* OR walk* OR strid* OR "physical activity"; (ii) absolute OR "absolute intensity" OR mets OR metabolic equivalent OR actigraph* OR acceleromet*; (iii) relative OR "relative intensity" OR "heart rate" OR "heart rate reserve" OR "vo2 reserve" OR vo2* OR "vo2 uptake" OR HRmax* OR metmax. Second, categories (i) to (iii) were combined using 'AND' and duplicates removed. Manual searching of the reference list of identified articles was also undertaken.

References were imported into Mendeley software (Mendeley desktop, Version 1.19.8, London, UK) for data management. After duplicates were removed, the titles and abstracts were screened independently based on the inclusion criteria by two reviewers (AW and either AB, GA, or NV) using Rayyan online software [29]. The full text of each potentially eligible study was retrieved and assessed independently by two reviewers (AW and either AB, GA, or NV) using the eligibility criteria. Discrepancies that could not be resolved by discussion were resolved by a third reviewer (AB or GA). Extracted data included: study setting, study

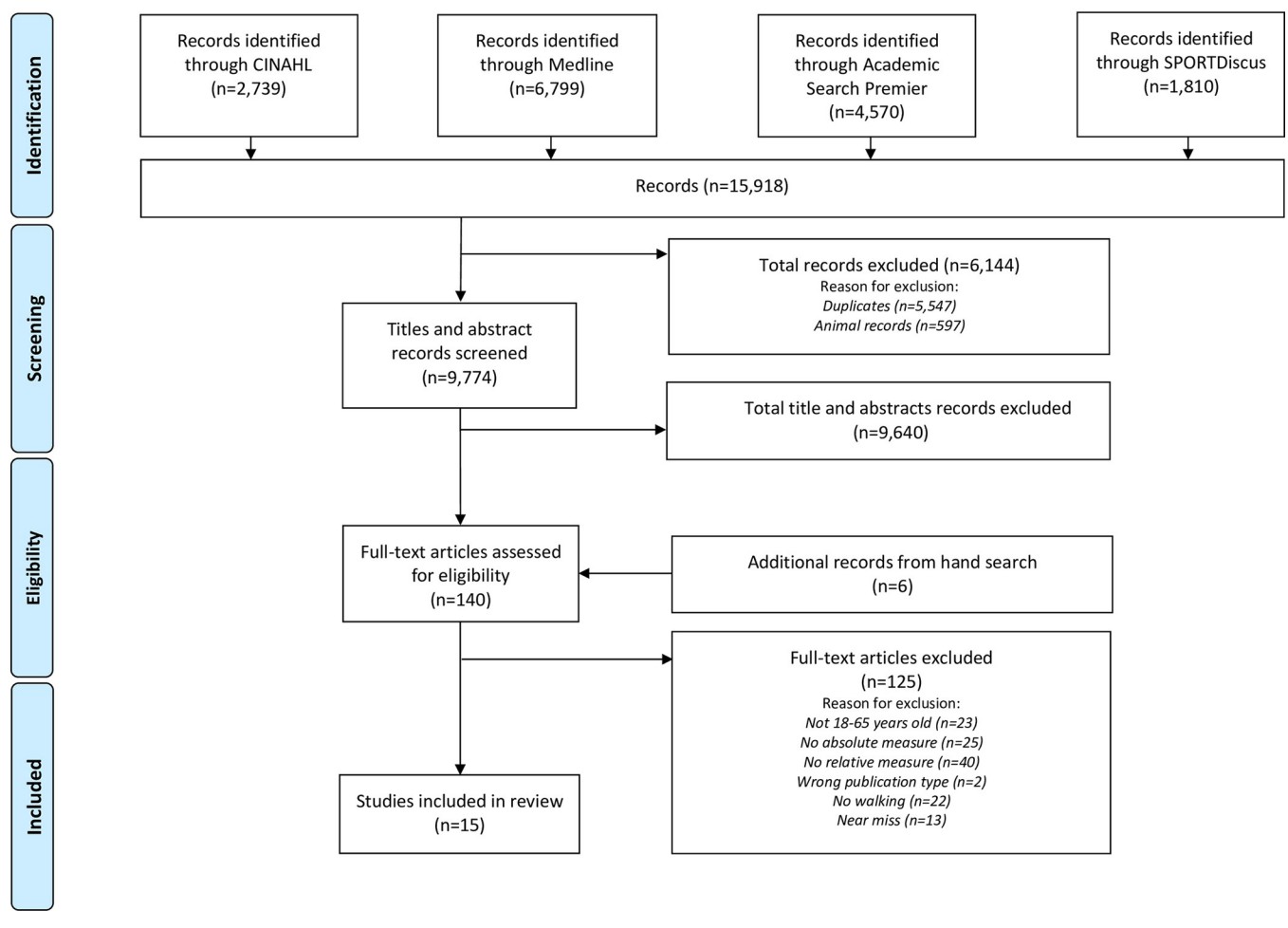

**Fig 1. PRISMA flowchart.**

population, participant demographics, relative intensity measure, absolute intensity measure and walking protocol. Missing data were requested from authors.

## Assessment of methodological quality

Quality assessment was evaluated by two authors (AW, GA), with a third author involved to resolve any disagreements (AB or NV). A modified Downs and Black scale [30] was used to rate quality in non-randomised controlled studies. The Downs and Black scale [30] was altered to incorporate questions relevant to the methodological design of the studies included. The modification of questions was completed by two reviewers and agreed by a third author, in line with other adapted models used (see S1 Table) in previously published systematic reviews and meta-analyses [31–33]. The Downs and Black scale [30] evaluates the quality of reporting, external validity, bias, confounding variables and power. Usually based on 27 criteria, the modified scale used a criterion of 13 items. Quality ratings were also adapted to provide relevant quality outcomes [34] (see S1 Table).

## Data collection

Raw data were requested from authors of all eligible studies included in this systematic review with meta-regression. Of all authors contacted, six did not respond to our communication, two authors no longer had the data available, and four were unable to supply additional data due to ethical reasons (no detail supplied). Where possible comparable (relative) variables [23, 35, 36] were amalgamated to create one large data set. Data were extracted from each eligible laboratory trial and stage. i.e., those that had completed a multistage incremental protocol, and provided relative and absolute values of intensity at each incremental stage.

## Data analysis

A narrative synthesis of the outcome measures and methodology was first undertaken. Data were also assessed for heterogeneity and a meta-regression conducted using a Bayesian random-effects approach [37]. There are several advantages to this approach over traditional methods: (1) the ability to estimate between-study heterogeneity along with its uncertainty, (2) allocate more weight to the results of particular types of study (e.g., randomised controlled trials), (3) provide exact likelihoods, (4) allow for uncertainty in all parameters, (5) allow for other sources of evidence (e.g., prior distributions), and (6) allow direct probability statements using different measurement scales [38]. The approach took the point estimate and standard error from individual studies and modelled these to produce an overall population Cohen's $d$ estimate for the effects of interest along with a measure of between-study variability. Data modelling was conducted using R [39]. The brms package [40] was used to perform this modelling, which is a package that uses Stan as the MCMC engine [41]. Subgroup analysis was conducted considering study design (e.g., experimental; randomised controlled trials, non-randomised controlled trials, intervention, observational and free-living design) as well as physical activity intensity–absolute (METs) and relative (%VO$_2$R %HRR, %HR$_{peak}$, %VO$_{2max}$, HR$_{index}$, lactate and ventilatory thresholds). A Bayesian regression was conducted to quantify the relationship between METs and %HRR, where a modelling approach was taken by fitting models with different response distributions and selecting and reporting the models with the best out of sample predictive performance using Leave-One-Out Cross Validation (LOO) to compare the expected log-predictive density (ELPD) of models [42]. We report population uncertainty as 95% credible intervals.

## Results

### Description of studies

A total of 15,918 papers were identified from the preliminary keyword search (see Fig 1). Prior to title and abstract screening, 6,144 records were excluded, including 5,547 duplicates and 597 animal studies. After title and abstract screening, 9,774 papers were eliminated based on the exclusion criteria, 6 additional papers were obtained from hand searches leaving 140 records for full-text review. The full-text review excluded 125 papers resulting in a total of 15 studies that met the inclusion criteria, [23, 35, 36, 43–54]. Common reasons for exclusion were participant age exceeded 65 years; exercise modality was not walking; and a direct comparison between a relative and absolute measure of physical activity was missing. All included studies reported at least one measure of absolute intensity (METs) and one relative measure of intensity (%HRR, %VO$_2$R, %HR$_{max}$, %VO$_{2max}$, HR$_{index}$, lactate threshold, ventilatory threshold). However, four studies were excluded from the assessment of agreement when based on the reported means [23, 43, 51, 54] because either, the absolute and relative values reported in these studies could not be directly compared using the ACSM intensity classification scheme [10] or mean and SD data were not reported. A further three studies were excluded from the assessment of agreement based on the reported means, either because the data from these studies were used in the comparison of agreement based on raw data [35, 36, 51] or used in the meta-regression [23, 35, 36, 51] This resulted in eight studies being examined for agreement when based on mean values [44–48, 50, 52, 53].

### Study quality

Studies were graded on a modified Downs and Black [30] scale, with a mean (SD) rating of 10.2 (1.2) from a maximum of 16, with a range of 7–12 (see S1 and S2 Tables).

### Systematic review

**Characteristics of study population.** Participant characteristics from the 15 studies included in this review are displayed in Table 1. Sample size ranged from 12–210 participants, with a mean (SD) sample size of 54 (48) participants. Participant age ranged from 25–61 years with a mean (SD) age of 38.8 (11.5) years. Sex was 70% female. Participant geographical location included United States (7 studies), Japan (2 studies), Spain (2 studies), Italy (1 study), Netherlands (1 study) Australia (1 study) and Brazil (1 study).

**Agreement between mean absolute and mean relative intensity.** Based on eight studies [44–47, 50, 52–54], a comparison of mean absolute (METs) and mean relative (%HRR, %HR$_{max}$, %VO$_2$R, %VO$_{2max}$, HR$_{index}$) intensity was conducted. 'Agreement' was operationally defined when both the mean absolute intensity and mean relative intensity values were classified as the same level of intensity using the exercise intensity guide published by the ACSM (Table 6.1 from [10]). For example, if mean METs was 4 and mean %HRR was 50%, then we classified this as 'agreed', because both values would be classified as 'moderate' intensity according to the guidelines. Whereas, if mean METs was 4 and mean %HRR was 35%, then this was classed as 'not agreed', because METs would be classified as 'moderate' intensity yet %HRR would be classified as 'light' intensity. From eight studies (n = 299 participants) there was 60% agreement between mean absolute and relative intensity categories when based on the ACSM guidelines for cardiorespiratory intensity (very light, light, moderate, vigorous or near maximal) (see Table 2).

**Agreement between raw absolute and raw relative intensity.** In addition to the examination of agreement between mean absolute and relative intensity, we also examined the

**Table 1. Characteristics of the studies included in the systematic review and meta-regression.**

| Study | Country | Mean (SD) age (years) | N/ sex | Absolute measure | Relative measure | HR$_{max}$ measured (direct or equation) | RHR measured (protocol/ direct) | VO$_{2max}$ (measured or estimated) | Measured or standard MET | Walking protocol | Protocol (single stage or multistage) | Walking speed (standardised or self-elected) |
|---|---|---|---|---|---|---|---|---|---|---|---|---|
| **Agiovlasitis et al.** [43] | United States | 51 (6) | 20 F 16 M | METs | HR$_{index}$ | N/A | 6 mins seated post 10 mins of sitting | N/A | Measured (seated) | 6x 6 min trials overground | Multistage | Standardised |
| **Tumiati et al.,** [44] | Italy | Intervention 48 (12) Control 51 (11) | 35 F 17 M | METs | %HRR | Equation | Not reported | Not reported | Standard | 2km·h⁻¹ brisk walk on indoor track | Single stage | Self-selected |
| **Caballero et al.** [45] | Japan | F 38.0 (11.7) M 39.5 (10.6) | 20 F 20 M | METs | %HRR | Equation 208 – (0.7xage) | 7-minute seated average taken every minute | N/A | Measured (seated 7 minutes) | 3 treadmill walking conditions | Multistage | Standardised |
| **Dos Anjos et al.** [36] | Brazil | 43.8 (20.2–64.9) | 121 F 89 M | METs | %HRR | Equation Gellish formula [13] measured | Measured, using a 40-minute supine protocol | N/A | Measured (seated 10 mins) | 6x3 min treadmill stages | Multistage | Standardised |
| **Ham et al.** [49] | USA | 31 (4.3) | 7 F 5 M | METs | %HRR | Equation Karvonen formula [16] | Not reported | N/A | 3 previously outlined accelerometer cut points | Ambulatory data | N/A | N/A |
| **Kilpatrick et al.** [46] | USA | 25.8 (7.9) | 9 F 11 M | METs | %HRR % VO$_2$R | Measured | Predicted | Estimated | standard | Walking performed at 2.5 mph for women 3.0 mph for men. incline increased by 2% every minute until an RPE of 13 was achieved. | Multistage | Standardised |
| **Nakanishi et al.** [47] | Japan | 38.65 (11.13) | 21 F 21 M | METs | %HRR | Equation Karvonen formula [16] | Measured 7 minutes seated | N/A | Standard | Overground laboratory trials using a pace leader | Single stage multiple speeds | Standardised |
| **Ozemek et al.** [48] | USA | F 25.7 (5.7) M 28.8 (7.2) | 35 F 38 M | METs | %HRR | Measured | Measured (protocol not reported) | Measured (submax 85% HR$_{max}$) | Standard | 5-minute incremental treadmill stages | Multistage | Standardised |
| **Sell et al.** [50] | USA | 24 (4) | 18 F 6 M | METs | %VO$_2$R | Measured | Measured 15 minute seated | Measured | Measured | 30-minute Treadmill | Single stage | Self-selected |
| **Gil-Rey et al.** [23] | Spain | Low fit 61.4 (5.8) High fit 56.6 (4) | 88 F 0 M | METs | LT | N/A | N/A | N/A | Measured | Incremental shuttle test | Multistage | Standardised |
| **Gil-Rey et al.** [51] | Spain | 57.2 (5.0) | 30 F 0 M | METs | %HRR % VO$_{2max}$ | Measured | N/A | Measured | Measured | overground walking trials | Multistage | Standardised |

*(Continued)*

Table 1. (Continued)

| Study | Country | Mean (SD) age (years) | N/ sex | Absolute measure | Relative measure | $HR_{max}$ measured (direct or equation) | RHR measured (protocol/ direct) | $VO_{2max}$ (measured or estimated) | Measured or standard MET | Walking protocol | Protocol (single stage or multistage) | Walking speed (standardised or self -elected) |
|---|---|---|---|---|---|---|---|---|---|---|---|---|
| **Hagins et al.** [52] | USA | 31.4 (8.3) | 18 F 2 M | METs | $\%HR_{max}$ | 208 - (0.7 x age) | N/A | N/A | Measured | 10-minute treadmill trials | Multistage | Standardised |
| **Brooks et al.** [53] | Australia | F 39.9 (2.8) M 40.0 (3.3) | 36 F 36 M | METs (predicted) | $\%HR_{max}$ | Predicted (208 – 0.7x age) | N/A | N/A | Measured | 15-minute overground walking | Single | Self-selected |
| **Spelman et al.** [54] | USA | 34.9 (8.6) | 22 F 7 M | METs | % $VO_{2max}$ $\%HR_{max}$ | Measured | No reported | Measured | Measured | 8 min treadmill trials | Single stage | Self-selected |
| **Sweegers et al.** [35] | Netherlands | 49.9 (9.0) | 50 F 0 M | METs | % $VO_{2peak}$ | Measured | Measured in supine position | Measured | Measured | 6 minutes treadmill trials | Multistage | Standardised |

**Table 2. Mean (SD) for measures of relative and absolute exercise intensity from each study included in the systematic review and meta-regression.**

| Author | Speed | Classification | Predicted METs | Measured METs | %HRR | %VO$_2$R | %HR$_{max}$ | % HR$_{peak}$ | %VO$_{2max}$ | % VO$_{2peak}$ | HR$_{index}$ |
|---|---|---|---|---|---|---|---|---|---|---|---|
| **Agiovlasitis et al. [43]** | 0.5 m.s$^{-1}$ | * | - | 2.23 (0.44) | - | - | - | - | - | - | 1.33 (0.11) |
| | 0.75 m.s$^{-1}$ | * | - | 2.59 (0.48) | - | - | - | - | - | - | 1.37 (0.14) |
| | 1 m.s$^{-1}$ | * | - | 2.96 (0.52) | - | - | - | - | - | - | 1.5 (0.14) |
| | 1.25 m.s$^{-1}$ | * | - | 3.53 (0.59) | - | - | - | - | - | - | 1.72 (0.12) |
| | 1.5 m.s$^{-1}$ | * | - | 4.43 (0.87) | - | - | - | - | - | - | 2.06 (0.31) |
| | Preferred walking speed | * | - | 4.07 (1.08) | - | - | - | - | - | - | 1.57 (0.24) |
| **Tumiati et al., [44]** | Self-selected pace 2 km WT | 1 | 3.8 (1.0) | | 55 (14) | | | | | | |
| | | 1 | 3.7 (1.2) | | 56 (13) | | | | | | |
| | | 1 | 4.4 (0.9) | | 59 (1.1) | | | | | | |
| | | 0 | 4.8 (1.3) | | 63 (11) | | | | | | |
| | | 1 | 3.5 (1.1) | | 54 (13) | | | | | | |
| | | 1 | 3.3 (0.6) | | 55 (13) | | | | | | |
| | | 1 | 3.4 (1.1) | | 52 (14) | | | | | | |
| | | 1 | 3.3 (1.3) | | 50 (19) | | | | | | |
| **Caballero et al. [45]** | 55 m.min$^{-1}$ | 0 | 3.3 (0.5) | - | 21.8 (5.3) | - | - | - | - | - | - |
| | 70 m.min$^{-1}$ | 0 | 3.7(0.5) | - | 26.0 (8.0) | - | - | - | - | - | - |
| | 100 m.min$^{-1}$ | 0 | 5.1 (0.9) | - | 36.5 (10.8) | - | - | - | - | - | - |
| | 13 m.min$^{-1}$ | 0 | 9.45 (1.5) | - | 73.4 (12.6) | - | - | - | - | - | - |
| **Kilpatrick et al. [46]** | Averages at 13 RPE | 0 | 5.8 (1.5) | - | 61.9 (16.9) | 46.8 (14.7) | 75.6 (10.5) | - | 51.5 (1.5) | - | - |
| | | 1 | | | | | | | | | |
| | | 1 | | | | | | | | | |
| | | 1 | | | | | | | | | |
| **Nakanishi et al. [47]** | 55 m.min$^{-1}$ | 0 | 3.35 (0.54) | - | 18.81 (9.1) | - | - | - | - | - | - |
| | 70 m.min$^{-1}$ | 0 | 3.75 (0.5) | - | 23.37 (9.14) | - | - | - | - | - | - |
| | 100 m.min$^{-1}$ | 0 | 5.12 (0.88) | - | 34.16 (11.87) | - | - | - | - | - | - |
| **Sell et al. [50]** | Self-selected brisk walk | 1 | | - | 4.8 (1.3) | - | 41.2 (3.5) | - | - | - | - | - |
| **Hagins et al. [52]** | 3.2 km.h$^{-1}$ | 0 | - | 2.5 (0.4) | - | - | 50.7 (8.0) | - | - | - | - |
| | 4.8 km.h$^{-1}$ | 1 | - | 3.3 (0.4) | - | - | 58.1 (10.7) | - | - | - | - |
| **Brooks et al. [53]** | Men—5.2 (0.6) | 1 | 3.76 (0.53) | | - | - | 52 (7) | - | - | - | — |
| | Women- 5.5 (0.5) | 1 | 4.1 (0.58) | - | - | - | 61 (9) | - | - | - | - |
| **Spelman et al. [54]** | Habitual walking speed | 1 | 5.1 (1.2) | - | - | - | - | - | 51.5 (1.2) | - | - |
| **Dos Anjos et al. [36]** | 1.11 m.s$^{-1}$ 0% | ** | 3.1 (0.03) | 3.8 (0.1) | 40.7 (1.3) | - | - | - | - | - | - |
| | 1.56 m.s$^{-1}$ 0% | ** | 4.0 (0.04) | 5.0 (0.1) | 56.0 (1.5) | - | - | - | - | - | - |
| | 1.56 m.s$^{-1}$ 2.5% | ** | 4.7 (0.04) | 5.8 (0.1) | 66.0 (1.9) | - | - | - | - | - | - |
| | 1.56 m.s$^{-1}$ 5% | ** | 5.3 (0.05) | 6.5 (0.1) | 72.3 (1.9) | - | - | - | - | - | - |
| | 1.56 m.s$^{-1}$ 7.5% | ** | 6.1 (0.06) | 7.3 (0.1) | 77.5 (1.6) | - | - | - | - | - | - |
| | 1.56 m.s$^{-1}$ 10% | ** | 6.9 (0.07) | 8.2 (0.1) | 81.7 (2.2) | - | - | - | - | - | - |

(*Continued*)

**Table 2.** (Continued)

| Author | Speed | Classification | Predicted METs | Measured METs | %HRR | %VO$_2$R | %HR$_{max}$ | % HR$_{peak}$ | %VO$_{2max}$ | % VO$_{2peak}$ | HR$_{index}$ |
|---|---|---|---|---|---|---|---|---|---|---|---|
| Sweegers et al. [35] | 3.2 km.h$^{-1}$ 0% | ** | - | 3.5 (0.9) | - | - | - | - | 51.0 (17.7) | - | - |
| | 4.8 km.h$^{-1}$ 0% | ** | - | 4.4 (0.9) | - | - | - | - | 60.8 (16.0) | - | - |
| | 5.5 km.h$^{-1}$ 5% | ** | - | 6.3 (1.3) | - | - | - | - | 83.8 (15.0) | - | - |
| Gil-Rey et al. [51] | - | $ | - | - | - | - | - | - | - | - | - |
| Gil-Rey et al. [23] | - | $ | - | - | - | - | - | - | - | - | - |
| Ham et al. [49] | - | $ | - | - | - | - | - | - | - | - | - |
| Ozemek et al. [48] | - | $ | - | - | - | - | - | - | - | - | - |

1 There is agreement between absolute and relative intensity

0 There is no agreement between absolute and relative intensity

*The absolute and relative values for this study cannot be compared because there is no guidance on how HR$_{index}$ can be classified into light, moderate and vigorous intensity thresholds.

** The absolute and relative values for this study cannot be compared as the raw values are compared in Fig 2.

$ The absolute and relative values for this study cannot be compared as the authors have not provided mean (SD) data for the walking trials

agreement between raw data from individual participants. Raw data were obtained from three authors, from four separate studies [23, 35, 36, 51]. Three studies had comparable relative intensity [35, 36, 51] variables and were compiled into one large data set, totalling 290 participants. The fourth study [23] was excluded from this analysis because there is no intensity classification for lactate threshold that enables comparison with the ACSM classification scheme, and therefore METs. A Bayesian random intercept logistic regression was conducted to examine the agreement between relative and absolute intensity (Fig 2), showing an agreement between absolute and relative intensity in 43% of all trials from these three studies.

**Relationship between absolute and relative intensity.** Two studies [36, 51] had identical relative variables (%HRR) and were compiled into a dataset containing 1257 individual data points from 240 participants. A series of Bayesian random intercept regression models were developed to quantify the relationship between METs and %HRR. The best performing model was a log-log regression model (Fig 3) which indicated that for every 1% increase in METs, % HRR increased by 1.12% (95% CI: 1.10–1.14). Specifically, the model predicts at the lower bound of absolute moderate intensity (3 METs), %HRR is 33% (95%CI: 18–57) and at vigorous intensity (6 METs) %HRR is 71% (38–100). Given that all studies in the meta-regression used measured METs (i.e., resting VO$_2$ and oxygen consumption during exercise were measured), our results likely reflect an optimal model fit between METs and %HRR, because estimated MET values between individuals would likely be similar given the standardisation of 1 MET (i.e., 3.5 mL.kg$^{-1}$.min$^{-1}$).

## Discussion

The aim of this systematic review with meta-regression was to examine the agreement and relationship between absolute and relative intensity when measured during walking. The main findings are: (1) a large disagreement exists between absolute and relative intensity across the

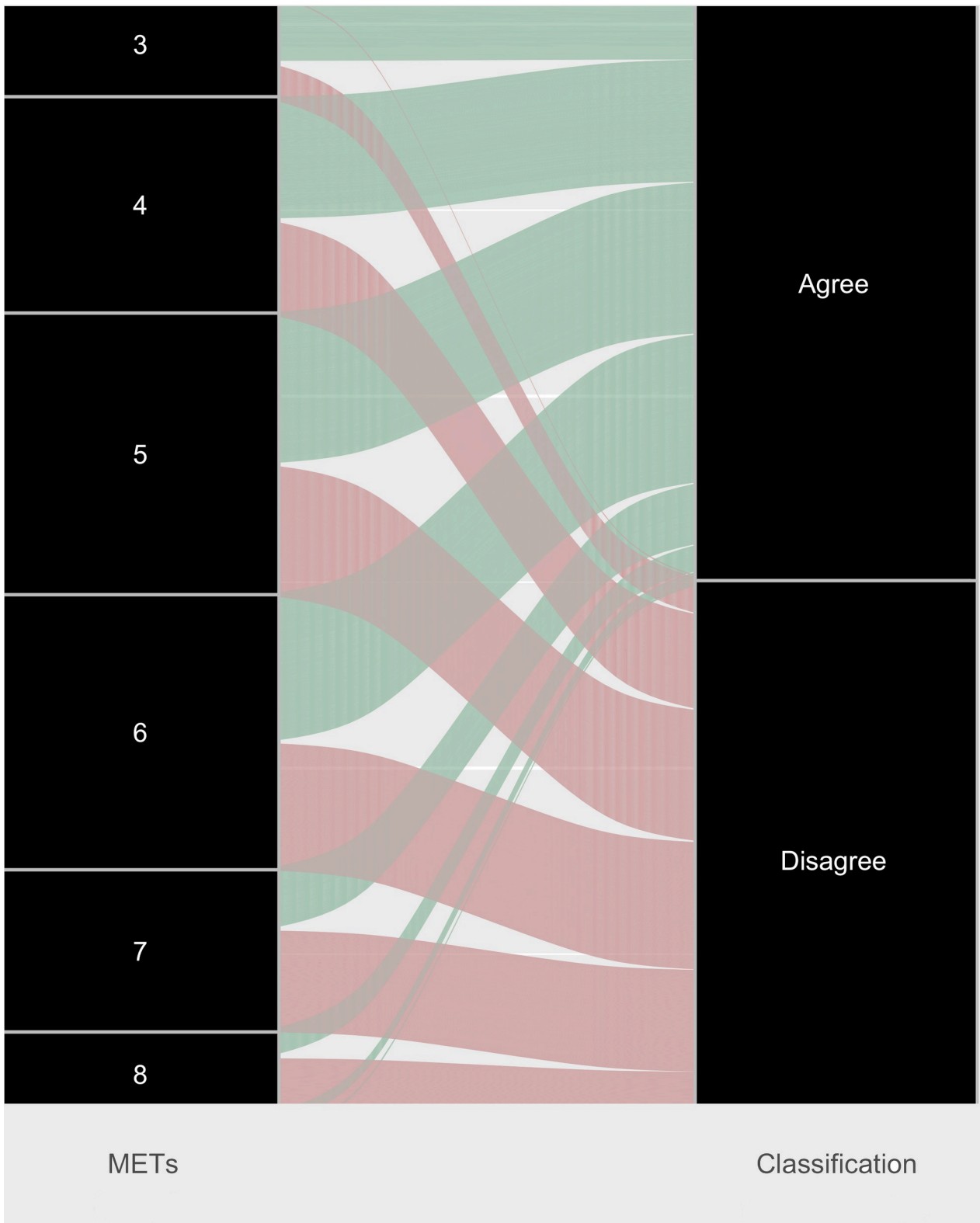

**Fig 2. The agreement between absolute and relative measures of moderate-to-vigorous exercise intensity when based on raw data from three studies [35, 36, 51].**

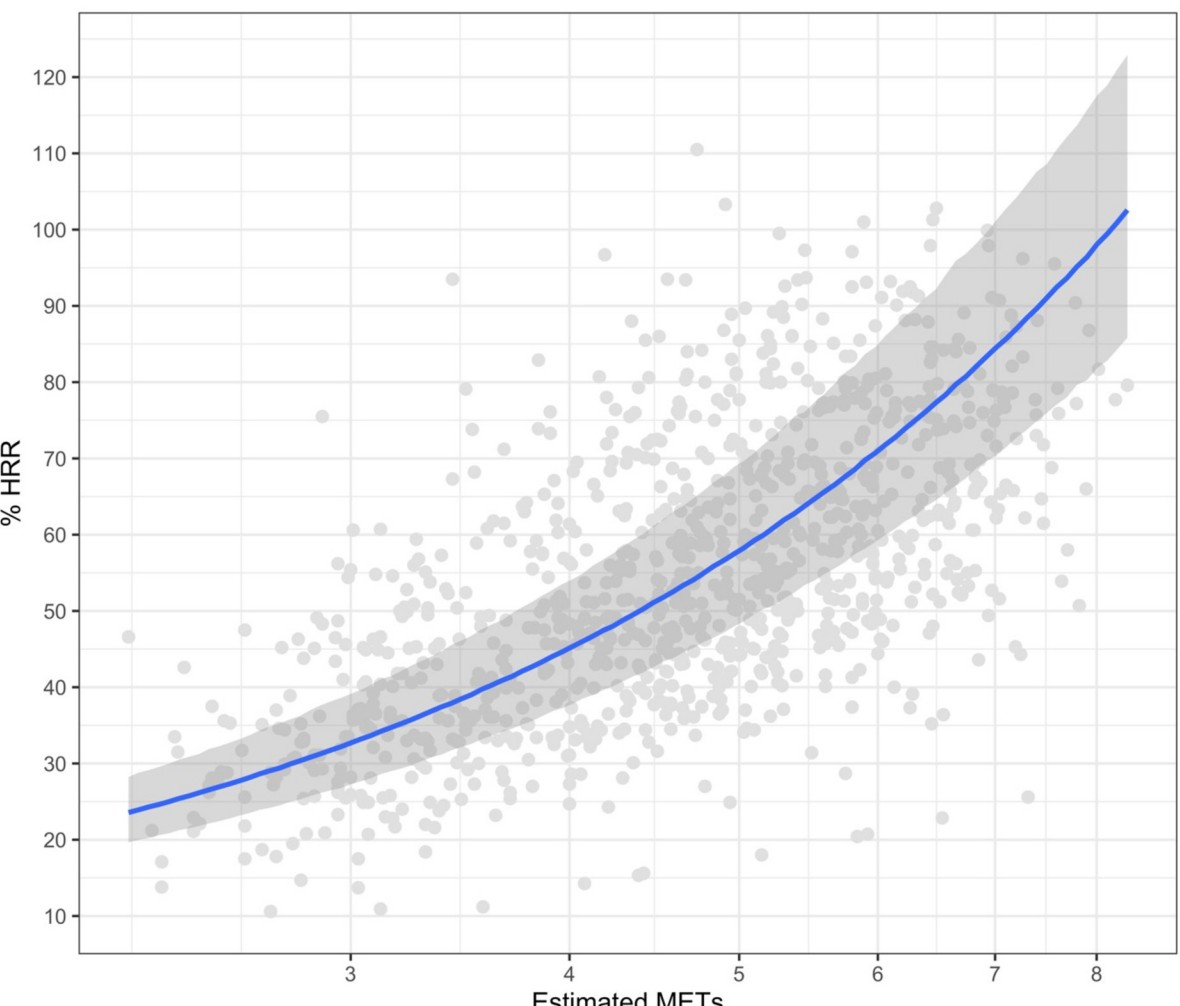

**Fig 3. Bayesian random intercept logistic regression.** The number of data points = 1257 due to each participant contributing multiple data points from multistage incremental tests. Please note that 5 data points for %HRR are above 100%, which is physiologically implausible. However, these are the data we obtained from the authors.

intensity spectrum; (2) the relationship between absolute and relative intensity suggests that METs overestimates moderate intensity; and (3) a large inter-individual variation exists in relative intensity for any given value of absolute intensity.

## Systematic review

**Study characteristics.** All 15 studies used METs as their absolute measure of intensity. However, several measures were used for relative intensity. The most common relative method was %HRR (8 studies) [36, 46, 48, 50–54], followed by %HR$_{max}$ (3 studies) [52–54], %VO$_{2max}$ / %VO$_{2peak}$ (3 studies) [23, 35, 46], %VO$_2$R (2 studies) [46, 50], lactate thresholds (2 studies) [23, 51], and HR$_{index}$ (1 study) [43]. This large variation in relative intensity methods may indicate why METs have become a popular choice among research and industry professionals as their use has been standardised as a measure of intensity.

**Study quality.** The quality score for included studies is reported in S1 Table. Eleven of fifteen studies included in this review scored 10 or above highlighting that, in general, the quality

of studies was acceptable. Question thirteen relates to the completion and reporting of power calculations. For a study to obtain four points for this criterion, *a priori* power calculation was required, with associated inputs reported and the correct number of participants recruited. All studies included in this review failed to meet any of this criterion, with none reporting a sufficiently completed *a priori* power calculation. This further highlights the need for reporting guidelines, such as STROBE, when designing and reporting observational studies. Adhering to guidelines and sufficiently powering studies is a key component of robust methodological design, and one that may significantly impact the validity and reliability of findings [55]. Question nine highlighted the need for pre-registration of studies prior to data collection and subsequent publication. Only two of the included studies pre-registered their methods and expected outcomes [36, 44]. There are a number of benefits of improving transparency of academic publishing, through the process of pre-registration, not least the reduction of bias and exaggeration of findings [56] and may also suppress or prevent p-hacking, HARKing and cherry picking [57] as hypotheses and analytical methods are declared prior to experimental trials being performed. Reducing the risk of bias within study design is fundamental to scientific rigour, and it is strongly recommended that future studies include sample size estimation and pre-registration to reduce risk of bias and improve the transparency of scientific publication.

**METs.**   There are discrepancies in how METs are measured, and three approaches were used in the included studies, estimated, hybrid, and measured METs. In its most absolute form, METs were estimated by using a standardised value (i.e., the same value for everybody) of 3.5 mL.kg$^{-1}$.min$^{-1}$ to prescribe and record intensities. This value was then multiplied to describe the intensity, which is a method that is traditionally used by accelerometers to develop velocity cut points [35, 48, 51–54]. That is, 3 METs (moderate intensity), should correspond to a $VO_2$ of approximately 10.5 mL.kg$^{-1}$.min$^{-1}$. However, there is evidence to suggest this standardised resting $VO_2$ value is not accurate for large portions of the population [58, 59], and could therefore contribute to overestimating moderate intensity. Five of the 15 studies in this review used this method for calculating METs [35, 44–46, 51].

METs can also be measured via a hybrid method incorporating breath-by-breath gas analysis, which is divided by a standardised value of 3.5 mL.kg$^{-1}$.min$^{-1}$ to provide a MET intensity. Two studies used this method [53, 54]. A third method firstly involved establishing an individualised MET value as measured by resting $VO_2$ from breath-by-breath gas analysis. Oxygen consumption was then measured directly during exercise. The exercise oxygen consumption was then divided by the individualised resting MET value to provide a fully individualised intensity categorisation. This method of determining METs was used in four studies [35, 43, 50, 52]. One consideration when measuring resting oxygen consumption (as used in the measured METs method) is that there is no criterion or consensus method for establishing resting $VO_2$. For instance, different protocol lengths (e.g. 6–40 minutes), body positions (e.g. supine vs seated), and data analysis methods (e.g. using the mean of all data collected vs the mean of the middle portion of the protocol) have been used [35, 43, 50, 52].

**Agreement between absolute and relative intensity methods.**   When examining the agreement between absolute and relative intensity, there were large disagreements between the two methods. When mean absolute and mean relative measures were compared in 8 of the 15 studies, there was 60% agreement. This indicated that a large proportion of absolute physical activity (METs) were classified differently when compared to relative measures. The disagreement between absolute and relative intensity was also highlighted in the larger raw data set, from data provided by three authors [35, 36, 51]. In this case, there was only a 43% agreement between absolute (METs) and relative measures (%HRR, %$VO_2$peak) (Fig 2). This clear disagreement between absolute and relative methods indicates large discrepancies when measuring intensity at an individual level. Clearly, an absolute measure of intensity such as METs is

participant independent. That is, the physiological capacity of individuals will not be captured by an absolute measure such as METs. Even in the studies using measured METs, and measured resting $VO_2$, the ability of a MET to capture the physiological response in an individual relative to their cardiorespiratory fitness is limited. The classification of intensity categories observed in our review has implications for both research and practice. For example, if moderate, vigorous physical activity (MVPA) is overestimated in individuals who are monitoring their physical activity based on METs, then the return on their physical activity investment will be lower than expected.

**Relationship between raw absolute and raw relative intensity.** The results of the Bayesian random intercept regression based on 240 participants and 1257 data points highlight that at 3 METs (absolute moderate intensity) the equivalent relative intensity is approximately 32%, substantially under the criterion 40% HRR required to meet moderate intensity reported in the ACSM guidelines [28]. However, the 95% credible interval of 18% to 57% emphasises the wide spread of relative intensity observed at 3 METs. This large variation in relative intensity at 3 METs implies that individuals could be performing relative physical activity spanning from very light intensity to the upper bound of moderate intensity when exercising at an absolute moderate intensity of 3 METs. At the lower bound of moderate intensity, the minimum intensity recommended for optimal health outcomes [60, 61], this sample were substantially under (10%) the required 40% minimum relative value. These findings suggest large inaccuracies in using absolute measures of physical activity intensity and provides further evidence that an individualised approach to physical activity intensity using relative measures should be considered.

From the Bayesian random intercept regression, it was only at 8 METs that all individuals' relative intensity was classified as moderate, reinforcing the large discrepancy associated with absolute measures of physical activity intensity. Using METs to prescribe and monitor exercise and physical activity has clear limitations within its measurement properties and has the potential to be misguiding for a large proportion of the population [58, 61, 62]. These results have important health implications. The inaccuracy of moderate intensity measurement, in addition to the large level of disagreement between intensity classifications, indicates the possibility of overestimation of total daily and weekly physical activity performed when using absolute measures.

The results of this systematic review with meta-regression indicate that measures of absolute and relative intensity often disagree, which is in line with previous findings [63, 64]. Consequently, health professionals should be aware of this lack of agreement between absolute and relative measures. Moreover, if we are to move towards a more individualised approach to exercise prescription and monitoring as advocated [65], relative intensity could be more highly prioritised. In the past, absolute intensity was preferred probably because the technology for measuring relative intensity was either not available or inappropriate. However, the expanding use of wearable devices by the wider population and the seismic growth in the wearable market [66] now offer the ability to use relative measures of intensity more easily. For example, the Apple Watch[TM] (the most popular smartwatch) now has over 100 million active users [67]. The use of wearable technology for measuring and guiding physical activity has been reported to improve the tailoring of exercise and physical activity to the individual and overall adherence [68, 69]. Evidence suggests that heart rate measured at rest and during walking by photoplethysmography sensors embedded in wearable devices is very accurate [70–72]. We also need more research on the use of relative intensity measures for long-term adherence and effectiveness of physical activity programmes.

**General considerations.** While this meta-regression has focussed on relative intensity that can be measured using a wearable device and are thus accessible to the general population, it is worth noting there are some limitations in doing this. Although %HRR and other measures of

relative intensity included in our study are individualised to a certain extent, they use fixed percentages to demarcate intensity categories (e.g., 40–59% HRR equals 'moderate' intensity). However, physiological markers such as ventilatory threshold are individualised to a greater extent because they reflect a real physiological threshold for each individual [63]. Moreover, when exercise intensity is anchored to individual ventilatory thresholds, it has been reported that the metabolic stimulus is better normalised across people with varying fitness levels [73]. However, wearable devices do not currently have the ability to measure such physiological thresholds and therefore %HRR may currently offer the most accurate relative measure available at a population level. A positive step forward, albeit a challenging one from a technological perspective, will be for manufacturers of wearable devices to incorporate sensors capable of measuring physiological markers such as ventilatory threshold. When this technological challenge has been met, we will have a fully individualised measure of relative intensity available at a population level.

One of the issues that needs to be considered is the range of MET and %HRR values included in our meta-regression. Given our focus on walking, the average %HRR value across the two studies included in the meta-regression was 50%, which would be classified as 'moderate' intensity [10], and might constrain the ability of the meta-regression to capture the agreement between absolute and relative intensity. However, as can be seen in Fig 3, a large proportion of data points are equal to or above 60% HRR, considered to be the lower bound of 'vigorous' intensity [10]. This suggests that although we focussed on walking our results do include a wide range of exercise intensities. Nevertheless, including studies involving running in a future meta-regression would allow a comparison of absolute and relative intensity over a wider range of exercise modes and intensities.

As well as whole-body responses to absolute and relative intensity exercise (e.g., oxygen consumption, METs), it is also important to consider responses at a cellular level. For example, mitochondria have a variety of important roles in both health and disease prevention [74, 75]. More specifically, relative exercise intensity, but not absolute exercise intensity, has been highlighted as an important determinant of exercise-induced changes that modulate early events of mitochondrial biogenesis [76]. As well as cellular responses, we also need to consider whether health outcomes are achieved by an absolute exercise intensity and thereby an absolute energy expenditure [77]. Although there is some evidence that an absolute energy expenditure of 2 MJ·wk$^{-1}$ is associated with an improvement in cardiovascular health [78], large errors in estimates of gross weekly energy expenditure are apparent [79], thereby casting doubt over the usefulness of both absolute exercise intensity and absolutely energy expenditure for achieving health outcomes.

## Conclusion

The conclusions drawn from this systematic review with meta-regression highlight the preferential use of METs as a method for quantifying absolute intensity in physical activity studies. However, we present strong evidence to suggest that the accuracy of METs on an inter-individual basis is not adequate for physical activity prescription and monitoring to maximise potential health benefits. As such, measurement of relative intensity could be more highly prioritised as part of physical activity programmes and guidelines and incorporated into wearable devices, to allow the wider population access to relative individualised intensity thresholds.

## Supporting information

**S1 Checklist. PRISMA 2020 checklist.**
(DOCX)

**S1 Table. The modified Downs and Black scale used to rate quality of included studies.**
(PDF)

**S2 Table. The modified Downs and Black score for each included study.**
(DOCX)

**S1 Appendix.**
(PDF)

## Acknowledgments

We would like to thank the authors who contributed raw data sets for the meta-regression.

## Author Contributions

**Conceptualization:** Ashley Warner, Natalie Vanicek, Amanda Benson, Grant Abt.

**Data curation:** Ashley Warner, Natalie Vanicek, Amanda Benson, Tony Myers, Grant Abt.

**Formal analysis:** Ashley Warner, Amanda Benson, Tony Myers, Grant Abt.

**Investigation:** Ashley Warner, Amanda Benson, Grant Abt.

**Methodology:** Ashley Warner, Natalie Vanicek, Amanda Benson, Grant Abt.

**Project administration:** Ashley Warner.

**Supervision:** Natalie Vanicek, Amanda Benson, Tony Myers, Grant Abt.

**Writing – original draft:** Ashley Warner.

**Writing – review & editing:** Natalie Vanicek, Amanda Benson, Tony Myers, Grant Abt.

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
