## [Decision Letter · Decision Letter 0]

14 Apr 2022

PONE-D-22-05506Agreement and relationship between measures of absolute and relative intensity during walking: a systematic review with meta-regressionPLOS ONE

Dear Dr. Abt,

Thank you for submitting your manuscript to PLOS ONE. After careful consideration, we feel that it has merit but does not fully meet PLOS ONE’s publication criteria as it currently stands. Therefore, we invite you to submit a revised version of the manuscript that addresses the points raised during the review process. As the authors will notice, there was strong agreement between reviewers in relation to the main criticisms to this manuscript. They both highlighted important omissions of key papers related to the main topic of this manuscript, and they also highlighted that the discussion of relative and absolute intensities should be better presented. Although the reviewers found merit in this manuscript, their overall perception was that some major re-crafting would be needed to increase the quality and potential reach of this manuscript. In general, I agree with the feedback that the reviewers provided and I believe that it should help the authors building a stronger story on a topic that is indeed relevant.

We look forward to receiving your revised manuscript.

Kind regards,

Juan M. Murias

Academic Editor

PLOS ONE

Journal Requirements:

2. Please provide the full results of the quality assessment for each study as a supplemental file.

3. Please include your tables as part of your main manuscript and remove the individual files. Please note that supplementary tables (should remain/ be uploaded) as separate "supporting information" files.

Reviewers' comments:

Reviewer's Responses to Questions

**Comments to the Author**

1. Is the manuscript technically sound, and do the data support the conclusions?

Reviewer #1: Yes

Reviewer #2: Yes

2. Has the statistical analysis been performed appropriately and rigorously? 

Reviewer #1: Yes

Reviewer #2: Yes

3. Have the authors made all data underlying the findings in their manuscript fully available?

Reviewer #1: Yes

Reviewer #2: Yes

4. Is the manuscript presented in an intelligible fashion and written in standard English?

Reviewer #1: Yes

Reviewer #2: Yes

5. Review Comments to the Author

Reviewer #1: Using a metanalytic approach, the present paper aimed to evaluate the discrepancy between METs-derived categories of intensity with indices of relative intensity that are deemed to delimitate those categories (e.g., %HHR, %HRmax, %VO2max, etc.). The analysis is interesting and, in general, I think it will contribute to the recent wealth of manuscripts advocating for the employment of more accurate methods for exercise intensity prescription/quantification. In summary, this analysis reveals that, an absolute intensity classification in many cases results in a mis-categorization of the actual “objective” intensity. Surely, however, this is something that has been thoroughly discussed in the past. See for example Norton et al. (2010, JSAMS) and Iannetta et al. (2021, Sports Medicine) – on a side note, it is surprising that these references were not included in this manuscript.

Another point that should be considered is that the authors assumed that maximum-derived percent indices of intensity (e.g., VO2max, HRmax) are the benchmark metrics against which METs levels should be compared. It is more and more evident that these metrics, although potentially improving the prescription of exercise compared to a METs-based approach, are still unable to capture the true metabolic stress of any physical activity (see Iannetta et al. 2020, MSSE). What the results of this meta-analysis demonstrate is a substantial, intrinsic incoherence between the METs-derived classification and relative metrics of intensity that are deemed to characterize those categories. Although I have no doubt that this incoherence has important practical implications, these results should not encourage the authors to imply that maximum-derived percent indices of intensity should be the “gold-standard” approach. In fairness, these limitations should also be discussed.

Finally, in several parts of the manuscript the authors give to the reader the impression that the METs system is often used to prescribe exercise intensity. In fact, what happens in most research and applied settings is that METs categories are used to retrospectively quantify/characterize the “training load” of physical activity, whereas they are rarely used to “prospectively” prescribe exercise intensity. This, I think, is an important distinction that the authors should consider implementing.

Specific comments:

The intro is quite long considering the objective of this meta-analysis. The authors could simply justify the need for this analysis based on several previous indications that the METs classification should be re-evaluated.

Data collection/Data analysis

It seems carefully conducted and carefully explained.

Results

Only 15 studies included. This is not the fault of the authors, but I wonder whether the same research question could have been answered more accurately with a more empirical methodology (actual collection of data).

Discussion

As for the intro, the discussion is quite long and, in some points, quite repetitive. I would suggest the authors to be more straight forward to avoid these repetitions.

Reviewer #2: This systematic review and metanalysis tested the agreement between measures of absolute and relative intensity during a specific form of exercise i.e. walking gait, in healthy adults.

The study found: i) a large disagreement between indexes of absolute and relative intensity measured contextually during walking; ii) a substantial overestimation of intensity when using METs; iii) a large inter-individual variability in relative intensity at a given value of absolute intensity.

The study has the merit to question the correspondence between absolute and relative intensity that is proposed in current guidelines, which is a very relevant and “hot” topic for the area of exercise prescription.

The study is well written and appears methodologically solid. However, I do not appreciate the utterly empirical approach and the lack of the physiological background that forms the very bases of the intensity prescription scheme. The authors are advised to broaden their focus to include recent pertinent work by Caen K and Teso M. for a physiological description of the concept of training intensity, the difference between absolute and relative intensity and why both these variables are important. Moreover, the authors should consider several recent papers from Iannetta D. These papers offer a physiology-based and data-driven critique of the fixed METs prescription scheme.

Major:

1- Walking is only mentioned in introduction while the authors don’t come back to it in the rest of the manuscript. Actually, the choice of restricting the focus of the paper on this form of exercise only seems unfortunate. When one considers the relatively low cost of walking locomotion (unless it is conducted on a steep slope or while carrying heavy loads) it is hard to generate anything but a moderate intensity of exercise. The validity of the study would have been higher if a wider range of absolute and relative intensities had been considered. Please discuss.

2- I am surprised of the relatively high absolute intensities reported in the study (>6 METs) as well as the high relative intensities (>60 %HRR). It is generally very hard to generate these absolute and relative intensities while walking. Please discuss.

3- The above leads to another question: were METs directly measured in the studies (based on measures of VO2) or were they guidelines-based estimates? This is very important and it should be clarified. If VO2 was not measured yours would be a rather circular argument. Please discuss.

4- To establish the agreement between indexes of absolute and relative intensity, my understanding is that you used the ACSM’s fixed classification scheme. However, the validity of this approach has been recently questioned (Sports Med. 2021 Nov;51(11):2411-2421; Med Sci Sports Exerc. 2020 Feb;52(2):466-473). This scheme is based on incorrect assumptions: i) the relative intensity is a linear function of absolute intensity, while clearly it is not since intensity domains are demarcated by threshold boundaries; ii) one size fits all in terms of mutual correspondence between absolute and relative intensity, which is clearly not the case due to differences in fitness and due to the higher than expected variability in the relative position of intensity boundaries. In the above papers a more accurate and fitness-dependant mutual translation scheme between relative and absolute intensity can be found. Please discuss all of the above.

5- I don’t agree with your conclusion that since there is a low correspondence between absolute and relative intensity (as classified by the ACSM’s translation scheme), we should only focus on relative intensity when prescribing exercise. I think that you study, using an alternative approach compared to the physiological studies, is further corroborating the idea that the current mutual translation scheme between absolute and relative intensity should be substantially revised.

However, this does not take away the importance of absolute intensity as an index of absolute energy expenditure, absolute heat production, absolute carbohydrates and fats oxidation, i.e. the “extensive” stress for the body that is generated by physical exertion. This is just as relevant towards certain health related outcomes of the “exercise pill” as the “intensive” stress that is related to the disturbance of the intracellular homeostasis in myocytes, that in turn is related to relative intensity. Absolute and relative load are both “active principles” of the exercise pill that stress the body in a different way, causing different structural and functional adaptations over time (please see J Physiol 2017 May 1;595(9):2915-2930 and successive work from the same group for a general framework).

6. PLOS authors have the option to publish the peer review history of their article (what does this mean?). If published, this will include your full peer review and any attached files.

Reviewer #1: No

Reviewer #2: **Yes: **Silvia Pogliaghi

---

## [Author Response · Author response to Decision Letter 0]

21 May 2022

Agreement and relationship between measures of absolute and relative intensity during walking: a systematic review with meta-regression

PONE-D-22-05506

Reviewer 1 comments Response to Reviewer Comments

Surely, however, this is something that has been thoroughly discussed in the past. See for example Norton et al. (2010, JSAMS) and Iannetta et al. (2021, Sports Medicine) – on a side note, it is surprising that these references were not included in this manuscript. 

Previous research has been completed in this area; however, we do not believe a systematic review with meta regression has. We acknowledge the suggested references and the Ianetta et al. 2021 study has been added (line 161 – 166)

Another point that should be considered is that the authors assumed that maximum-derived percent indices of intensity (e.g., VO2max, HRmax) are the benchmark metrics against which METs levels should be compared. It is more and more evident that these metrics, although potentially improving the prescription of exercise compared to a METs-based approach, are still unable to capture the true metabolic stress of any physical activity (see Iannetta et al. 2020, MSSE). What the results of this meta-analysis demonstrate is a substantial, intrinsic incoherence between the METs-derived classification and relative metrics of intensity that are deemed to characterize those categories. Although I have no doubt that this incoherence has important practical implications, these results should not encourage the authors to imply that maximum-derived percent indices of intensity should be the “gold-standard” approach. In fairness, these limitations should also be discussed.

While we acknowledge that the relative measures used within this paper are not gold standard measures, we do not state these relative measures to be gold standard. Currently, wearable devices are limited to measuring heart rate (and not ventilatory or lactate thresholds) and thus at a population level offer the most accurate method of relative exercise intensity. To clarify this, the following has been added in lines 161-166:

‘Yet there are some limitations with this method when compared to other forms of relative intensity such as lactate and ventilatory thresholds that do not use a fixed percentage value for exercise intensity boundaries (1). However, wearable devices do not currently have the capacity to measure such physiological thresholds and currently, %HRR may offer the most accurate relative measure available at a population level.’

This has also been referred to on lines 495 – 504 

In the past, absolute intensity was preferred probably because the technology for measuring relative intensity was either not available or inappropriate. However, the expanding use of wearable devices by the wider population and the seismic growth in the wearable market (85) now offer the ability to use relative measures of intensity more easily. Evidence suggests that heart rate measured at rest and during walking by photoplethysmography sensors embedded in wearable devices is very accurate (86–88). Individualised physical activity relative to cardiorespiratory fitness is now more readily available, due to wearable devices, and therefore, physical activity and prescription should adopt relative intensity measures as part of population-based guidelines.

Finally, in several parts of the manuscript the authors give to the reader the impression that the METs system is often used to prescribe exercise intensity. In fact, what happens in most research and applied settings is that METs categories are used to retrospectively quantify/characterize the “training load” of physical activity, whereas they are rarely used to “prospectively” prescribe exercise intensity. This, I think, is an important distinction that the authors should consider implementing.

This was not our intention. Our intention was to describe physical activity monitoring. The manuscript has been amended to reflect this on line 464 and 486 we refer to exercise monitoring and on lines 504 and 513 to ‘prescription and monitoring’. We believe there is importance in physical activity guidelines being prescribed as accurately as possible and thus refute the notion that exercise prescription should not use relative formats, especially when wearable devices make such information readily accessible.

Specific comments:

The intro is quite long considering the objective of this meta-analysis. The authors could simply justify the need for this analysis based on several previous indications that the METs classification should be re-evaluated. 

We have adjusted the introduction to reflect this whilst adding additional references to justify the rationale.

Only 15 studies included. This is not the fault of the authors, but I wonder whether the same research question could have been answered more accurately with a more empirical methodology (actual collection of data). 

We agree that more empirical studies in this area are needed, but the purpose of a systematic review is to synthesise the findings of studies currently available in order to then be able to design appropriate methodology for future studies. A single study is not likely to have the sample size required to do as proposed. Moreover, the meta-regression approach has allowed us to examine the relationship between METs and %HRR in a large sample that would have been very difficult to achieve with a single study. 

Discussion

As for the intro, the discussion is quite long and, in some points, quite repetitive. I would suggest the authors to be more straight forward to avoid these repetitions. 

As suggested, we have re-evaluated the discussion and removed text accordingly so that it is more succinct.

Reviewer 2 comments 

I do not appreciate the utterly empirical approach and the lack of the physiological background that forms the very bases of the intensity prescription scheme. The authors are advised to broaden their focus to include recent pertinent work by Caen K and Teso M. for a physiological description of the concept of training intensity, the difference between absolute and relative intensity and why both these variables are important. Moreover, the authors should consider several recent papers from Iannetta D. These papers offer a physiology-based and data-driven critique of the fixed METs prescription scheme.

Whilst we acknowledge the empirical approach, physiological mechanisms were not the focus of this systematic review with meta-regression. We acknowledge the Ianetta et al. 2021 paper recommended and have added this to the introduction: line 161-166.

We have also highlighted specifically why we chose to utilise the relative measures due to their population level accessibility in the conclusion line 496 – 505

In the past, absolute intensity was preferred probably because the technology for measuring relative intensity was either not available or inappropriate. However, the expanding use of wearable devices by the wider population and the seismic growth in the wearable market (85) now offer the ability to use relative measures of intensity more easily. Evidence suggests that heart rate measured at rest and during walking by photoplethysmography sensors embedded in wearable devices is very accurate (86–88). Individualised physical activity relative to cardiorespiratory fitness is now more readily available, due to wearable devices, and therefore, physical activity and prescription should adopt relative intensity measures as part of population-based guidelines.

Walking is only mentioned in introduction while the authors don’t come back to it in the rest of the manuscript. Actually, the choice of restricting the focus of the paper on this form of exercise only seems unfortunate. When one considers the relatively low cost of walking locomotion (unless it is conducted on a steep slope or while carrying heavy loads) it is hard to generate anything but a moderate intensity of exercise. The validity of the study would have been higher if a wider range of absolute and relative intensities had been considered. Please discuss.

While we acknowledge other modalities of physical activity exist, we specifically chose walking as we are interested in the largest population level effect, of which walking is the most common form of physical activity performed. This is stated in the introduction:

“Walking is a very popular form of physical activity at a population level (1). It is low cost, accessible, and well-tolerated across age groups (2). Given this popularity, walking is a key intervention for physical activity promotion (3). Advocating walking (and physical activity in general) is important because it is well documented that physical activity reduces the risk of developing a range of chronic diseases (4,5).”

In the discussion we also reference walking on lines 390 and 501

Although we acknowledge that walking usually generates moderate intensity, this is the aim of the paper, to understand how walking intensity is measured during and the relationship between relative and absolute intensity. At a population level physical activity is not adhered to well, and we believe that the largest effect on public health would be to encourage sedentary populations into doing more moderate intensity physical activity. Hence the advocation of moderate relative intensity as per the physical activity guidelines and the use of wearable devices within this paper.

I am surprised of the relatively high absolute intensities reported in the study (>6 METs) as well as the high relative intensities (>60 %HRR). It is generally very hard to generate these absolute and relative intensities while walking. Please discuss. 

Although we acknowledge walking generally elicits moderate intensity, it is possible, depending on the cardiorespiratory fitness of the individual, to achieve vigorous intensity during flat, non-loaded walking conditions. Additionally, the aim of the paper was to identify any differences between absolute and relative intensities at moderate intensity, in line with the minimum requirement intensity often prescribed by government guidelines. (England, Australia, Canada, USA all advocate for moderate intensity as the minimum intensity in exercise guidelines).

The above leads to another question: were METs directly measured in the studies (based on measures of VO2) or were they guidelines-based estimates? This is very important, and it should be clarified. If VO2 was not measured yours would be a rather circular argument. Please discuss. 

This has been discussed in the manuscript - please refer to the section entitled “METS’ line 428 – 446 for clarity. 

To establish the agreement between indexes of absolute and relative intensity, my understanding is that you used the ACSM’s fixed classification scheme. However, the validity of this approach has been recently questioned (Sports Med. 2021 Nov;51(11):2411-2421; Med Sci Sports Exerc. 2020 Feb;52(2):466-473). This scheme is based on incorrect assumptions: i) the relative intensity is a linear function of absolute intensity, while clearly it is not since intensity domains are defined by threshold boundaries; ii) one size fits all in terms of mutual correspondence between absolute and relative intensity, which is clearly not the case due to differences in fitness and due to the higher than expected variability in the relative position of intensity boundaries. In the above papers a more accurate and fitness-dependant mutual translation scheme between relative and absolute intensity can be found. Please discuss all of the above. 

Although we acknowledge this, the ACSM suggest %HRR is a gauge of relative intensity in line with METs, %VO2max, %VO2R etc, and currently is used as a one size fits all approach at a population level to prescribe and monitor physical activity intensity. While we appreciate and have acknowledged that other potentially more accurate methods of relative intensity monitoring exist (line 162 – 166), there is currently no feasible method of measuring/monitoring these physiological thresholds easily at a consumer/population level, hence our focus on variables that can easily be obtained from currently available wearable devices.

I don’t agree with your conclusion that since there is a low correspondence between absolute and relative intensity (as classified by the ACSM’s translation scheme), we should only focus on relative intensity when prescribing exercise. I think that you study, using an alternative approach compared to the physiological studies, is further corroborating the idea that the current mutual translation scheme between absolute and relative intensity should be substantially revised. However, this does not take away the importance of absolute intensity as an index of absolute energy expenditure, absolute heat production, absolute carbohydrates and fats oxidation, i.e. the “extensive” stress for the body that is generated by physical exertion. This is just as relevant towards certain health related outcomes of the “exercise pill” as the “intensive” stress that is related to the disturbance of the intracellular homeostasis in myocytes, that in turn is related to relative intensity. Absolute and relative load are both “active principles” of the exercise pill that stress the body in a different way, causing different structural and functional adaptations over time (please see J Physiol 2017 May 1;595(9):2915-2930 and successive work from the same group for a general framework). 

We have not made this conclusion, nor are we questioning the importance of absolute forms of energy expenditure, heat production or fat and carbohydrate oxidation. We conclude that measures of exercise prescription and monitoring should be more highly prioritised to improve access at a population-level, not lab-based.

‘As such, measurement of relative intensity should be more highly prioritised as part of physical activity programmes and guidelines’….. ‘to allow the wider population access to relative individualised intensity thresholds.’

Lines 514– 517.

---

## [Decision Letter · Decision Letter 1]

22 Jul 2022

PONE-D-22-05506R1Agreement and relationship between measures of absolute and relative intensity during walking: a systematic review with meta-regressionPLOS ONE

Dear Dr. Abt,

Thank you for submitting your manuscript to PLOS ONE. After careful consideration, we feel that it has merit but does not fully meet PLOS ONE’s publication criteria as it currently stands. Therefore, we invite you to submit a revised version of the manuscript that addresses the points raised during the review process. As the authors will see, the reviewers still find value in this manuscript. However, they both indicated that previous comments were only partly addressed and that the updated version still contains important conceptual limitations and physiological interpretations that need to be addressed properly. This tasks requires that the authors focus on making some major adjustments to the manuscript. After examining the manuscript and the comments from the reviewers, I am convinced that the feedback provided below offers the authors the opportunity to improve the quality of the work that they are presenting. I would encourage the authors to take advantage of the feedback they are receiving from experts on this topic.

We look forward to receiving your revised manuscript.

Kind regards,

Juan M. Murias

Academic Editor

PLOS ONE

Reviewers' comments:

Reviewer's Responses to Questions

**Comments to the Author**

1. If the authors have adequately addressed your comments raised in a previous round of review and you feel that this manuscript is now acceptable for publication, you may indicate that here to bypass the “Comments to the Author” section, enter your conflict of interest statement in the “Confidential to Editor” section, and submit your "Accept" recommendation.

Reviewer #1: (No Response)

Reviewer #2: All comments have been addressed

2. Is the manuscript technically sound, and do the data support the conclusions?

Reviewer #1: Partly

Reviewer #2: Partly

3. Has the statistical analysis been performed appropriately and rigorously? 

Reviewer #1: Yes

Reviewer #2: Yes

4. Have the authors made all data underlying the findings in their manuscript fully available?

Reviewer #1: Yes

Reviewer #2: Yes

5. Is the manuscript presented in an intelligible fashion and written in standard English?

Reviewer #1: Yes

Reviewer #2: Yes

6. Review Comments to the Author

Reviewer #1: General comments:

I want to begin this second round of review acknowledging that I think the paper is well written and it addresses a valid research question. It is unfortunate, however, that some, if not most, of my previous comments (as well as some valid ones from reviewer #2) were not taken into consideration. Also, some changes (for example those presumably made to shorten the intro and discussion) are not explained in the responses nor tracked within the updated manuscript version.

I’m still of the opinion that the authors are trying to “overinterpret” their data. Yes, there is a discrepancy between METs categories and relative intensity ones. Yet, I think the authors should be careful in overemphasizing this discrepancy and dismiss the use of METs. Afterall, from such a metanalytic approach one cannot derive whether this discrepancy is potentially detrimental for the characterization of exercise intensity (considering the limitations inherent to current relative intensity metrics).

Specific comments:

The point in relation to the validity of the relative intensity construct should be elaborated in the discussion instead of the intro (line 161-166). The assumption made throughout the manuscript that exercising at the same relative percentage of HRmax (or any other maximum derived percentage) equals the same level of relative intensity between two individuals is incorrect. Although I might agree that using a percent approach is likely more accurate than using METs, these limitations should be well elaborated within the body of the discussion.

Line 457: this is a bit unclear. The “intensity” associated with a given MET is participant dependent.

Line 494-496: statements like this one make it sound as if this is the first piece of evidence demonstrating this discrepancy – which is not the case (again see Norton et al. discussing this issue quite few years ago).

Line 502: Measures of relative intensities are already included, for example, in the ACSM guidelines. They are technically already adopted. The problem is that they are also limited.

In general, the authors make strong statements in relation to the adoption of relative measures of intensity. But which one? %HRmax or %HRR? Or %VO2max? Not necessarily they coincide either (see recent work from Ferri Marini et al.). Then, how should the general public derive these measures? In my opinion, there are several physiological and methodological limitations with any of these propositions. I just would like the authors to recognize some of them to offer a more thorough view of the problem of exercise quantification and prescription (which, I agree with the authors, is an important one in our field).

Reviewer #2: The authors have not adequately responded to the questions and criticism.

The way in which the authors reported questions and answers (i.e. incomplete reviewer’s text; homogeneous format between question and answer; referral to manuscript lines rather than explicit response to the reviewer question) does not facilitate the evaluation of the author’s response.

There is more to answering the criticism and suggestions of peers than just citing literature. Meta-analysis is a powerful tool to answer questions. However, unless an appropriate mechanistic context is provided, there is the risk to pose the wrong question as well as to misinterpret the results, which I think is the case for this manuscript. Appropriate physiological background needs to be in the paper.

As stated in the first revision, the study has the merit to question the correspondence between absolute and relative intensity that is proposed in current guidelines, which is a very relevant and “hot” topic for the area of exercise prescription. In strictly practical terms, the use of indexes of relative intensity rather than the (WRONG) MET scheme may be an easy-fix of the relatively complex problem of the implementation of exercise prescriptions. However, the correct approach to the matter is the physiology-based and data-driven revision of the fixed METs prescription scheme. This approach exists and has been published (Iannetta’s work as an example), it is valid across sexes and a very large age-range and it is clearly applicable outside laboratories. Moreover, a number of methods exist that allow an accurate estimate of absolute metabolic intensity based on the simple measure of HR in a variety of real-life contexts and populations (e.g. Wicks J, Pogliaghi S). These studies provide a broader context and support to your work and should be aknowledged and discussed.

The limitation of using only walking, i.e. an activity that elicits at best a moderate intensity, is hardly suitable to establish a correspondence between methods of measurement of absolute and relative intensity. A wider range should be used. The implications of using a limited intensity range must be discussed by the authors.

More importantly, it is clear from the authors answer that in the majority of the studies included in the meta-analysis, actual METS were not directly measured but rather they were estimates. The authors need to discuss the implications of this on the lack of correspondence between absolute and relative exercise intensity.

The revised version of your manuscript clearly states that the low correspondence between absolute and relative intensity (as classified by the ACSM’s translation scheme) should lead us to focus on relative intensity only when prescribing exercise in the general population. Again, this may appear an easy fix but it is mechanistically incorrect. What your analysis confirms that the current mutual translation scheme between absolute and relative intensity should be substantially revised. However, the importance of absolute intensity as an index of absolute energy expenditure, absolute heat production, absolute carbohydrates and fats oxidation, i.e. the “extensive” stress for the body that is generated by physical exertion should not be downplayed. This is just as relevant towards certain health related outcomes of the “exercise pill” as the “intensive” stress that is related to the disturbance of the intracellular homeostasis in myocytes, that in turn is related to relative intensity. Absolute and relative load are both “active principles” of the exercise pill that stress the body in a different way, causing different structural and functional adaptations over time (please see J Physiol 2017 May 1;595(9):2915-2930 and successive work from the same group for a general framework). This needs to be clearly presented in the manuscript.

7. PLOS authors have the option to publish the peer review history of their article (what does this mean?). If published, this will include your full peer review and any attached files.

Reviewer #1: No

Reviewer #2: **Yes: **Silvia Pogliaghi

---

## [Author Response · Author response to Decision Letter 1]

29 Aug 2022

Please see attached file for our responses to reviewer comments.

---

## [Decision Letter · Decision Letter 2]

19 Oct 2022

Agreement and relationship between measures of absolute and relative intensity during walking: a systematic review with meta-regression

PONE-D-22-05506R2

Dear Dr. Abt,

We’re pleased to inform you that your manuscript has been judged scientifically suitable for publication and will be formally accepted for publication once it meets all outstanding technical requirements.

I apologize for the delay in submitting this editorial decision, but I was waiting for final comment from one of the reviewers to be submitted. Given the delays, and considering the feedback from the other reviewer and my evaluation of your updated version, I decided to proceed without further delays in the process.

Kind regards,

Juan M. Murias

Academic Editor

PLOS ONE

Additional Editor Comments (optional):

Reviewers' comments:

Reviewer's Responses to Questions

**Comments to the Author**

1. If the authors have adequately addressed your comments raised in a previous round of review and you feel that this manuscript is now acceptable for publication, you may indicate that here to bypass the “Comments to the Author” section, enter your conflict of interest statement in the “Confidential to Editor” section, and submit your "Accept" recommendation.

Reviewer #1: All comments have been addressed

2. Is the manuscript technically sound, and do the data support the conclusions?

Reviewer #1: Yes

3. Has the statistical analysis been performed appropriately and rigorously? 

Reviewer #1: Yes

4. Have the authors made all data underlying the findings in their manuscript fully available?

Reviewer #1: Yes

5. Is the manuscript presented in an intelligible fashion and written in standard English?

Reviewer #1: Yes

6. Review Comments to the Author

Reviewer #1: I thank you the authors for considering my comments. They have addressed my concerns. I don't have any more comments.

7. PLOS authors have the option to publish the peer review history of their article (what does this mean?). If published, this will include your full peer review and any attached files.

Reviewer #1: No

---

## [Editor Report · Acceptance letter]

27 Oct 2022

PONE-D-22-05506R2 

Agreement and relationship between measures of absolute and relative intensity during walking: a systematic review with meta-regression 

Dear Dr. Abt:

I'm pleased to inform you that your manuscript has been deemed suitable for publication in PLOS ONE. Congratulations! Your manuscript is now with our production department. 

Kind regards, 

on behalf of

Dr. Juan M. Murias 

Academic Editor

PLOS ONE